# Sustainable Conservation and Reuse of Historical City Center Applied Study on Jeddah—Saudi Arabia

**Mohamed Ibrahim El-belkasy \* and Sahl Abdullah Wahieb**

Department of Engineering and Applied Science, Applied College, Umm Al-Qura University, Makkah Al-Mukarramah 24382, Saudi Arabia; sawahieb@uqu.edu.sa
\* Correspondence: mielbelkasy@uqu.edu.sa; Tel.: +966-554-968-912

**Abstract:** Developing a historical city center leads to city resilience and sustainable heritage conservation; changing the social fabric is a fundamental problem that affects historical and heritage areas. As a result of Jeddah's development, the community of the heritage area changed to another without the same interest or appreciation for the region's heritage. After listing it as a world heritage, the government is taking care of many conservation projects for Historic Jeddah. The research aims to evaluate a proposed project to reverse the last decade's social changes of the local community as a step in achieving historic center sustainability. The scope of this proposal is to select adaptive reuse of the listed historic buildings and provide the area with state-of-the-art services. This proposed project will attract the targeted community to return, which will achieve the research purpose. The research followed three different methodologies; through a theoretical approach, Jeddah city urban development and growth were highlighted, and the currently enforced conservation policies regulating land use were introduced. Moreover, the analytical approach studies the possibility of social reform of the local community by implementing adaptive reuse for heritage buildings. Three sustainability pillars were tested through a survey of three different stockholders. The research discussed the various stockholders' visions toward the aimed sustainable objectives. Hence, the applied part of the research evaluates the sustainability of the proposed project. The study finds that the local community is essential in the conservation process's sustainability. Reusing the heritage buildings in the resettled original community will sustain the conservation process and increase the real estate value of the Jeddah historical areas.

**Keywords:** heritage conservation; society replacement; heritage building reuse; heritage investment





## 1. Introduction

According to urban development, the historical center was exposed to social and economic changes, which changed the city center. The social and economic changes affect the historical center's land use, leading to changes in the urban fabric and visual image of these areas. The sustainability of heritage conservation in historical areas of the city depends on the reuse of heritage buildings and revival of the historical center's urban fabrics; therefore, a rehabilitation plan should use all of the urban fabric in this historical center [1].

Jeddah's historical city is considered the most important in the Saudi Arabia kingdom, and it is the most livable world heritage site in it. Historical Jeddah was suffering from deterioration due to the emergence of the oil era and rapid urban development of the city. This development changed Jeddah's city center, led to a change in historical city center land use, and displaced the original inhabitants to new urban extensions of the city, which led to the deterioration of the state of the historical heart of the city. The replacement of the population with another community was made up of non-Saudi expatriates. This change in the region's societal structure and land use led to many problems reflected in the area's urbanization [2]. The Kingdom has been interested in conserving the historical

heart of the city of Jeddah. Many attempts have emerged to develop policies to preserve the region since the beginning of the 1980s. The Kingdom has been interested in protecting the historical centers of cities since the beginning of the twenty-first century [3]. The Saudi Commission of Tourism and National Heritage, in cooperation with the Municipality of Jeddah, has developed policies to preserve the heart of the city's historical preparation for inscription on the World Heritage List, which was already placed on it in 2014.

Reusing heritage buildings and reintegrating them back into the life of the local community is one of the essential methods of sustaining the preservation of heritage areas [4]. Indeed, the preservation policies of Historic Jeddah have provided multiple alternatives to repurpose some of the important buildings on the historical site. Still, many buildings remain that have not been completed. Hence, the research idea is to reuse these buildings as residential buildings and re-attract Saudi society from the middle and upper-middle classes to reside in the region.

### 1.1. Research Goals

The research mainly aims to study the proposal to resettle the local community of Historic Jeddah that was displaced from it as a result of urban development in the city and to return it to the region. Another aim is to determine the impact of this proposal on the sustainability of conservation operations in the Jeddah heritage area and the extent of the proposal's compatibility with development projects in the region.

### 1.2. Research Hypothesis

The research hypothesis is that reusing heritage buildings in historic Jeddah for resettlement of former residents from the middle and upper-middle classes will help sustain the conservation process and provide new uses for restored buildings. The economic value of real estate will be raised and investments in services provided to the target community will be attracted, in addition to their compatibility with development projects in the region.

## 2. Materials and Methods

The research is divided into three stages. First, a literature review is performed on the study's theoretical method. The second stage is the analytical analysis of heritage conservation and sustainable building reuse factors. The last step depends on the practical approach used to evaluate the proposed project.

### Research Methodology

The research methodology is divided into three approaches; the first depends on the theoretical background and contextual approach. Studies of the policies of historical Jeddah conservation, the general plan of the region, studying the current situation of land use, the condition of the local community, and studying and reuse of heritage buildings in terms of the research proposal. The second approach depends on the analytical approach that deals with analyzing the potentials of historical Jeddah in terms of resettling layers of Saudi society in the historical area. The analytical side studies the elements of sustainability of reuse projects and determines the evaluation criteria for the project through sustainability elements. In contrast, the last applied approach evaluates the research proposal through the evaluation elements inferred from sustainability factors. The research will be evaluated through an investigation directed to a group of stakeholders and participants in the conservation process and the area's vacant to evaluate the proposal and reach the results and recommendations. Figure 1 shows research three approaches.

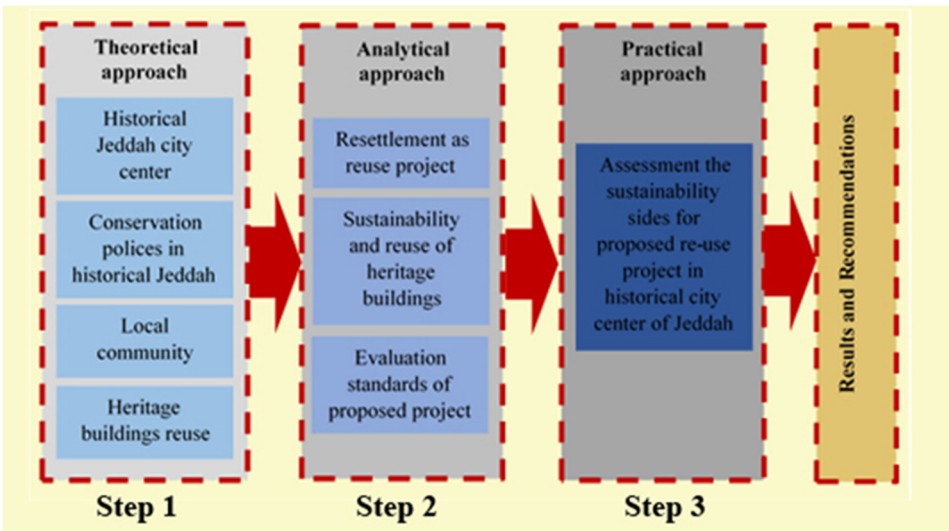

**Figure 1.** The research methodology.

## 3. Theoretical Background and Contextual Approach: Historical Jeddah City Center

Historic Jeddah was placed on the World Heritage List by UNESCO in 2014 [5]. The region is characterized by a distinct urban heritage that shows the characteristics of the architecture of the Hejaz region, which appears in the most important cities in Jeddah, Makkah, Al-Madinah, and Al-Taif. This character was formed due to an essential exchange of human values, technical know-how, building materials, and techniques across the Red Sea region and along the Indian Ocean routes between the 16th and the early 20th centuries as shown in Figure 2. Migration began from the old central area to the areas of new urban extensions, which led to the neglect of the historical area and replaced its residents with lower social classes and different nationalities [6].

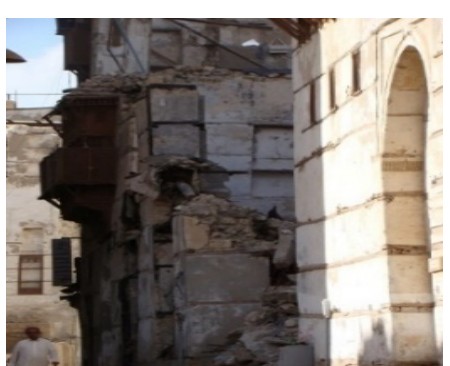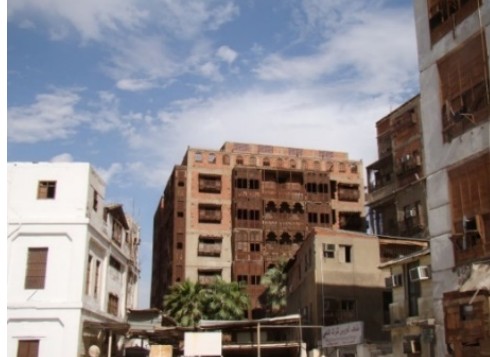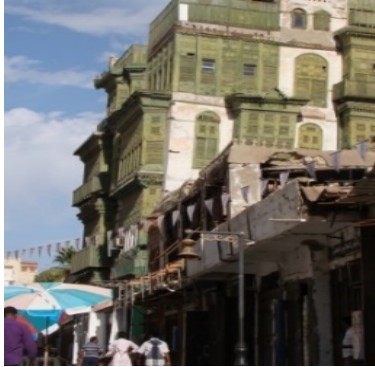

**Figure 2.** The architectural characteristics of historical Jeddah.

### 3.1. Conservation Policies in Historical Jeddah

There were many projects to preserve the historical area of Jeddah starting from the seventies and eighties of the last century, and these policies can be summarized in Robert Matthew's policies and Jeddah municipality policies.

### 3.1.1. Robert Matthew's Policies

Robert Matthew's policies depended on conserving the city of Jeddah so that the buildings of a distinctive character in the area are preserved, linking the heritage area with areas of new urban extensions, developing and raising the efficiency of the heritage area [7] see Figure 3, and achieving these goals was divided into two levels:

- The urban fabric: The area was divided into four heritage zones, the first of which is from the western side, in which the area was linked to new urban extensions, and services were provided. Its urban fabric is being changed.
- Classification of heritage buildings: Heritage buildings, especially in the third region, were classified into three categories (A-B-C) according to their historical importance and condition [8].

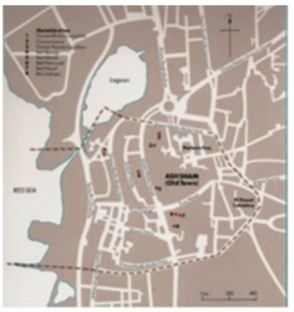 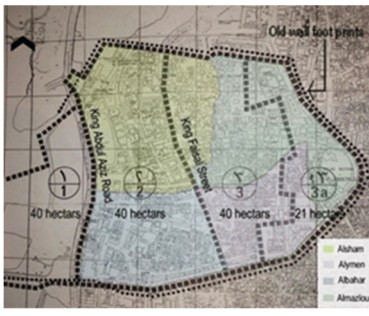 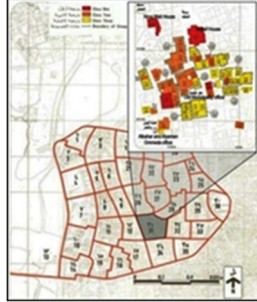

**Figure 3.** Robert Matthew's conservation policy for historical Jeddah [7].

### 3.1.2. Jeddah Municipality's Policies

The Municipality of Jeddah, in cooperation with the Saudi Commission of Tourism and National Heritage, has developed a set of policies for preserving Historic Jeddah to place the area on the World Heritage List. This is summarized in the following [9]:

- Prepare a list of significant buildings that must be preserved.
- Encourage owners of heritage homes to restore their homes under the supervision of the Commission to ensure the results of restoration operations.
- Preserve the original urban fabric of the area while providing appropriate landscape elements to the heritage and historical position of the site.
- Provide suitable parking spaces in the area and spreading heritage awareness among the local community. One of the most critical applications for achieving this goal was the work of heritage festivals.

### 3.2. *The Local Community of Jeddah's Historical City Center*

The diversity distinguished the Hijaz region in the society that comprises its cities due to Hajj and Umrah, which led to the arrival of many nationalities to the region, which helped integrate these different cultures and created a distinctive character in social and cultural terms. The diversity is evident in its architectural character, as well as in many social, economic, and cultural aspects. See Figure 4.

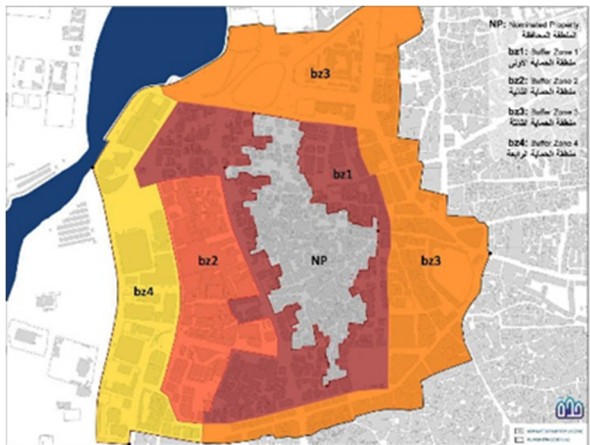 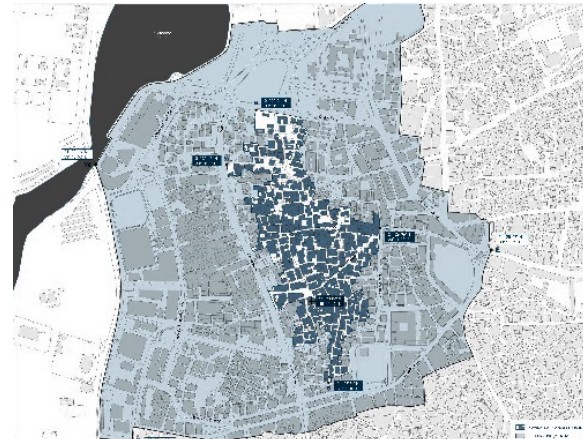

**Figure 4.** Historical Jeddah limits and the buffer zones of the heritage area [10].

The urban development that accompanied the emergence of oil in the Kingdom helped to attract the local community in the region to the new urban development areas, which led to:

- Replacing the local community of the area with another community made up of immigrants.
- Attention to commercial activities in the region at the expense of other activities.
- Residential activities were limited to some expatriate workers, and the available residential spaces in the residences were used as storage spaces.

These factors led to the deterioration of the area and the deterioration of the condition of many buildings and public spaces in the region, and then strategies for dealing with the region came to restore the buildings and restore the role of urban spaces in the region, which was applied in some celebrations and heritage festivals, for example, the Historic Jeddah Festival [8].

*3.3. Heritage Building Reuse as a Tool of Sustainable Heritage Conservation*

The sustainability of heritage conservation depends on the development of three main factors in the heritage environment: the local community, the economy, and the local environment [11]. The development processes should deal with the sustainability level at which the building must be dealt with so that the intervention does not depend on restoration. Here, the question must arise whether the building will continue to perform its role for which it was established, or will it perform a new function, and how will the building perform its new function so that a balance occurs between the three factors of sustainable development [12].

3.3.1. Reuse Heritage Buildings in Resettlement (Proposed Project)

Historic Jeddah contains many heritage buildings (537 buildings) [13], often used as homes. The reuse of heritage buildings must consider the ability of buildings to meet the needs of the new use. For example, compatibility between the new function and the spaces of the building, which applies to resettlement, in which the building used in the same historical job with making some modifications to accommodate the spaces with the mechanical requirements (elevators–air conditions) and technological (the Internet) abilities that the resettlement process may require according to the target class. Hence, the resettlement process is appropriate to the nature of the area and the nature of the buildings in it, which reduces the total cost of the reuse of these buildings [14], starting from restoration to resettlement, due to the lack of interventions required until the reuse process takes place. See Figure 5.

3.3.2. Historical Jeddah Potentials and Resettlement

Historic Jeddah possesses many capabilities that help the success of the reuse of heritage building projects in resettlement, and the most important of these capabilities are the following:

- The area contains many residential buildings and mixed-use buildings (according to Jeddah development planning), reducing the economic cost of reuse, which is spent on rehabilitating the spaces to suit the new function of the building. This will not be required as a result of reusing the building in the same function for which it was built.
- At present, the region represents a magnet for all investors and the local community, especially after the development projects targeting the region launched by His Highness, the Crown Prince of the Kingdom of Saudi Arabia, including, for example, the Heart of Jeddah project.
- The area possesses many commercial and cultural services [9], especially after the completion of development work for the area (center of Jeddah project) [15], which will represent attractions for the Saudi community of various classes to re-house and return to the area again.
- The areas surrounding the historic area, which represent the urban environmental buffer of the heritage area, can be developed and expanded the scope of resettlement

projects to increase the economic return from the project and produce new local architecture characterized by authenticity and contemporary to represent scope for protecting the heritage area and increasing the local community's awareness of the importance of architectural heritage and its role in producing future architecture that preserves the values, traditions, and cultural reference of the local community.

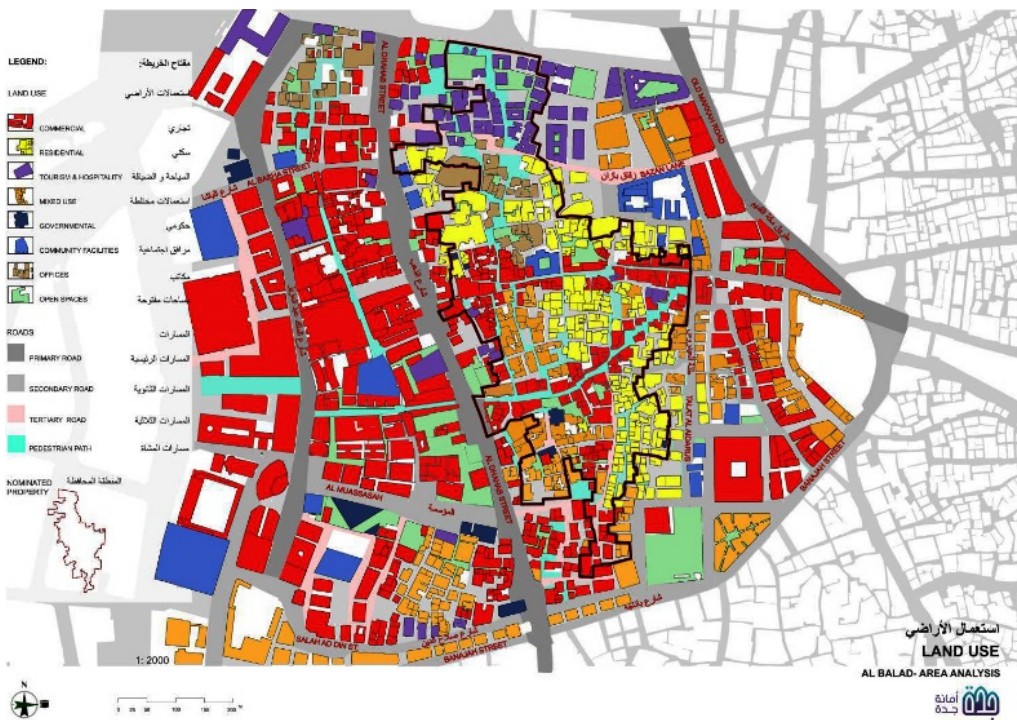

**Figure 5.** Historical Jeddah land use [10].

## 4. Analytical Approach: Reuse and Sustainability in Historical Jeddah Buildings

Historic Jeddah contains many heritage buildings. Historic Jeddah represents a heritage area or an integrated heritage neighborhood. To ensure the sustainability of the preservation operations of this neighborhood, the reuse of these buildings is one of the essential requirements of the sustainability process [16]. The Municipality of Jeddah has prepared lists to restore and repurpose a group of buildings, including, for example, five buildings that will be converted as museums; this is the importance of reusing the rest of the buildings, and where the historical Jeddah represents a model of an integrated heritage residential neighborhood, restoring the residential buildings to their old role is one of the solutions to reuse these buildings, and the sustainability of conservation operations in the area, the following must be available:

- Providing the necessary services to the population in the event of resettlement, considering the level of the population in the provision of services.
- Considering privacy, mainly because the area contains many commercial activities.
- Providing services related to the expected tourism activities in the region, especially after the completion of the Downtown Jeddah project, will put the region on the international tourism list, as planned.
- Attention to investment projects in the region that are commensurate with their nature and with the local community of the region to provide the necessary investments for the sustainability of conservation and reuse projects in the region.

## 5. Public Participation as a Tool to Evaluate the Sustainability of the Proposed Project

Public participation is one of the essential elements of the success and sustainability of conservation projects [17]. Therefore, ensuring the approval of the local community and its

partnership with the project is one of the most critical factors for its success and acceptance, especially since the properties in Historic Jeddah are private. The owners' acceptance of the project is an essential step in its implementation. The acceptance of Jeddah's local community with the idea of returning to the region is one of the most critical factors that can assess the sustainability of the project. In addition to all of the specialists, the local administration, and those in charge of conservation operations in the region, because they are the ones who must provide studies related to services and public spaces in the project and present them to the community, which represents an element among the elements of the project's attraction, from the above, we find that the evaluation should address all participants in the proposed project to show the project's ability to be sustainable and accepted by all parties of the reuse process.

## 6. Sustainability Levels in the Resettlement Project in Historical Jeddah

The sustainability of conservation and reuse of heritage buildings and areas operates on three overlapping levels: the local community of heritage areas, the economics of conservation and reuse projects, and the local environment [18]. It was conserving the heritage of the region and its integration with future projects that serve the city center of Jeddah and represent the urban extensions of the historic area.

### 6.1. Historical Jeddah Community

When dealing with the local community of the heritage area in Jeddah, we must differentiate between the current community of the region and the target community. The current community of the region is a community that has most expatriate workers and owners of the commercial activities that exploit the ground floors of buildings in commercial activities [19], in addition to exploiting the upper floors of buildings for storage or housing expatriate workers and engaging in commercial activities in the region.

The target community of the resettlement project, where project aims to attract the old local community of the heritage area of the upper and middle class from the people of a new city to change the combined structure of the area to return to its previous era, while providing the services required for the people of the area after studying the services that development projects in central Jeddah will provide. Its impact on the characteristics of the target community and the regional environment must be studied.

6.1.1. Factors Affecting the Historical Jeddah Target Community

The resettlement project aims to replace the current community of the area with the original community that left it previously as a result of changing the city center as previously. For this community to return to the region, it is necessary to study its current characteristics and the factors affecting it and its way of life so that it represents an attraction point to return people to the region.

- Privacy is one of the most critical factors affecting the target community for resettlement and the Saudi community in general, which must be considered, especially with the spread of projects that attract tourism in the region.
- As a result of the high level of per capita income in the Kingdom, the means of living on which families depend have changed and have become the basics of living, which must be considered when planning the area and for resettlement, such as parking spaces, traffic separation, and cars entering the area, because the current lifestyle of the community depends on it.
- One of the essential factors that characterize the part of Saudi society targeted for resettlement operations is family cohesion, which requires the study of the possibility of accommodating more than one family in one house. Maintaining their privacy and targeting families at the beginning of formation requires the study of the spaces needed for continued communication with the rest of the family.

6.1.2. Evaluating Criteria for Social Aspects

Social aspects can be assessed by studying the following factors:

- Local community acceptance of the idea of resettlement.
- Service efficiency is needed to cover the basic needs of the local community after resettlement.
- The appropriateness of the spaces in the heritage houses to the requirements of the target community.
- Urban spaces are appropriate for the lifestyle of the target community.
- The relationship between development projects in the region and resettlement projects.

*6.2. Economics and Investment of Historical Jeddah*

Architectural and urban heritage contains many cultural, aesthetic, and social values, whose economic value is difficult to estimate according to market requirements. Economic returns can be defined as the society-wide gains from the project compared to the situation that would prevail if the project were not undertaken [20]. There are many methods by which the economic value of heritage buildings is evaluated, for example, by values of direct use, values of indirect use, and values unrelated to use [21], and through these methods, the returns of heritage preservation projects and then the investment risks of these projects can be calculated [22].

Historic Jeddah contains 537 [13] (according to Robert Matthew's study) buildings classified as heritage buildings that represent real estate wealth in the case of direct use as residential buildings that play the same role for which they were established. Therefore, in this case, the evaluation will be by estimating the market value of these buildings by direct use-value. Property plays an essential role in conservation processes and reuse. The values of heritage properties tend to be more resistant during downturns than in the general market [23], so the investment of this property can change the governing framework of conservation projects [24]. In Historic Jeddah, the property represents one of the most critical factors affecting the reuse and investment of these buildings [6], although the government provided the necessary financial support for the restoration [9].

6.2.1. Financial of the Conservation Projects in Historical Jeddah

Conservation projects in Historic Jeddah are financed by the Kingdom represented by the Ministry of Culture and the Jeddah Municipality. The financing is through loans granted by the government to owners for restoring their homes under the supervision of the Ministry and the municipality [25] and special grants offered as a grant from His Highness the Crown Prince, to restore 56 homes, estimated at 50 million Saudi riyals [26]. Some self-efforts are working on restoring the property of the region, and the state covers the costs of region-wide conservation projects for squares and roads.

6.2.2. Evaluating Criteria for Economic Aspects

Economic aspects can be assessed by studying the following factors:

- The expected real estate value of the property before and after the implementation of the project.
- Encouraging investment in reuse projects and service projects for the target community.
- The economic impact of the project on tourism development.
- Providing job opportunities in the project and beyond in the population and tourism services in the region.

*6.3. Historical Jeddah Environment*

The research deals with the historical environment of Jeddah on three levels: the first level is the building design, the second one is the urban context, and the third level deals with changes in the heritage environment as a result of the conservation project.

### 6.3.1. Building Design

The architectural design of the heritage buildings in Historic Jeddah is compatible with the local environment (Figure 6) in terms of building materials compatibility and their ability to reduce the heat load of the building, natural ventilation of the internal spaces with the treatment of facades, and the use of the ratios of longitudinal windows and Roshan [27] to reduce the impact of solar radiation on the facades.

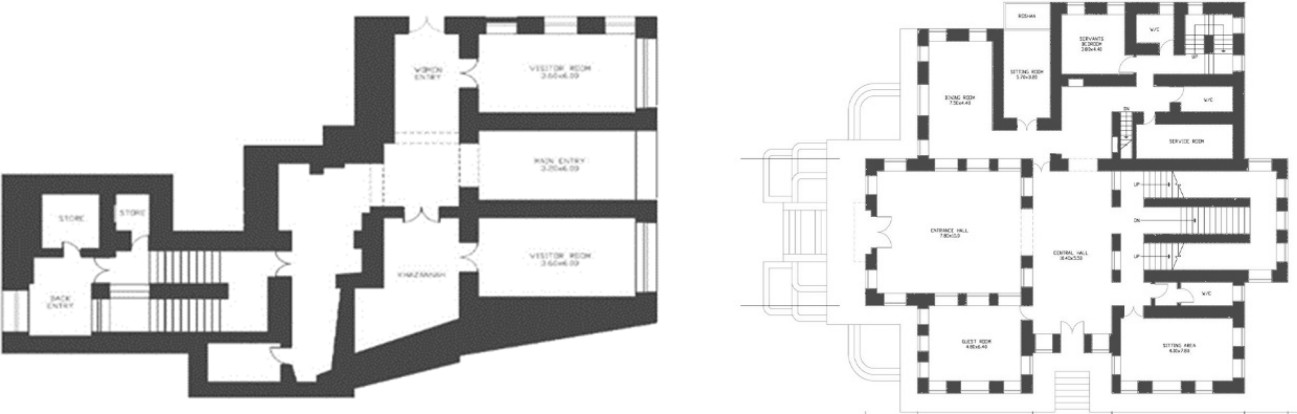

**Figure 6.** Examples of historical Jeddah house ground floor plan [3].

### 6.3.2. Urban Context

The urban fabric of Historic Jeddah expresses the heritage of Arab city that is compatible with the environment and represents a model of the sustainable urban fabric in terms of compact buildings' fabric [28] and proportions of streets and alleys that provide semi-shaded paths in extensive periods of the day and reduce the exposure of the external facades of buildings to solar radiation. Educational, recreational, and daily services are available in a nearby range that encourages its residents to walk and reduce their dependence on vehicles [12].

### 6.3.3. Conservation Project

The Historic Jeddah conservation project relied on improving the environment in the area, starting with restoring buildings and improving their condition by improving and paving roads and pedestrian paths with appropriate materials that match the site character. Additionally, landscape elements were provided, such as street lighting units, signboards, and garbage collection boxes, following the area's character [19]. These policies helped put the area on the World Heritage List in 2014.

### 6.3.4. Evaluating Criteria for Environmental Aspects

Environmental aspects can be assessed by studying the following factors:

- Renovation of existing heritage buildings and improvement of the visual image.
- Control pollution levels in the area and prevent the movement of vehicles.
- The appropriateness of residential use of heritage buildings to the environment and energy consumption control.
- The efficiency of the current cleaning and firefighting system will not be sufficient in the case of implementing the proposed project.

## 7. Practical Approach: Resettlement Project Assessment

The practical approach depends on investigation directed to stockholders, local community, specialists, and municipality to test the research hypothesis and the experimental sample is shown in Tables 1 and 2.

**Table 1.** Number, percentage, and distribution of participants.

|  |  | Frequency | Percent | Valid Percent | Cumulative Percent |
|---|---|---|---|---|---|
| Valid | local community | 42 | 63.6 | 63.6 | 63.6 |
|  | municipality | 8 | 12.1 | 12.1 | 75.8 |
|  | specialist | 16 | 24.2 | 24.2 | 100.0 |
|  | Total | 66 | 100.0 | 100.0 |  |

**Table 2.** The mean and standard deviation of acceptance of living in heritage buildings.

| Participant | Mean | N | Std. Deviation |
|---|---|---|---|
| local community | 3.0952 | 42 | 1.16472 |
| municipality | 2.8750 | 8 | 1.45774 |
| specialist | 2.3125 | 16 | 1.13835 |
| Total | 2.8788 | 66 | 1.22179 |

*7.1. Assessment of Social Aspects*

Assessment of social aspects depends on measuring the social factors (acceptance of living in heritage buildings, services efficiency, urban spaces and target community, relation between reuse project and development projects) as dependent factors according to the participant point of view as an independent factor. In the following assessment, we will discuss the result of the investigation. The main indicator result will be included in the research and other indicators will be in the Appendix A (Tables A1–A18).

7.1.1. Acceptance of Living in Heritage Buildings

According to previous tables, the participant accepts living in a heritage building that can be ratiocinated from the values of significance, which is more than 0.05, and the mean, which has close values (from 2.3 to 3; Table 3). The local community accepts the project more than the municipality and specialist, which means appreciation from the local community to the historical Jeddah. If the local community thinks of historical Jeddah as a residential district, it will help them to accept the resettlement project.

**Table 3.** ANOVA table for the previous factors.

|  |  |  | Sum of Squares | df | Mean Square | F | Sig. |
|---|---|---|---|---|---|---|---|
| accept living in HB relation with participant | Between Groups | (Combined) | 7.099 | 2 | 3.549 | 2.486 | 0.091 |
|  | Within Groups |  | 89.932 | 63 | 1.427 |  |  |
|  | Total |  | 97.030 | 65 |  |  |  |

7.1.2. Service Efficiency

According to the investigation, the participant mentions that the services in historical Jeddah are not enough for the community of the hypothesis project, which can be ratiocinated from the frequency of service efficiency (Table 4) and values of significance, which is more than 0.05, and the mean, which has close values of more than 3.6, meaning that all participants see that the recent service needs development as suitable for the proposed project.

**Table 4.** Mean and standard deviation of heritage buildings' reuse impact on the economic value of the property.

| Participants | Mean | N | Std. Deviation |
|---|---|---|---|
| local community | 1.8571 | 42 | 0.89909 |
| municipality | 1.7500 | 8 | 1.03510 |
| specialist | 1.7500 | 16 | 0.68313 |
| Total | 1.8182 | 66 | 0.85771 |

7.1.3. Evaluate the Appropriateness of Heritage Building Spaces for the Target Community

According to the investigation, the participant mentions that the services in historical Jeddah are not enough to the community of the hypothesis project, which can be ratiocinated from the frequency of the services efficiency (Table 4) and values of significance, which is more than 0.05, and the mean has close values of more than 3.5.

7.1.4. Evaluate Urban Space's Appropriateness to the Lifestyle of the Target Community

According to the investigation, the participants mention that the urban spaces in historical Jeddah are appropriate to target the community of the hypothesis project. However, the result reflects that the acceptance was not significantly different; the mean was 2.7, and the significance was 0.285 more than 0.05.

7.1.5. Evaluate the Relationship between the Resettlement Project and Development Projects

According to the investigation, the participant mentions a positive impact of the development project and the proposed project that was clear from the sample result; the mean was 2.3 and significance was 0.473.

*7.2. Assessment of Economic Aspects*

The assessment of economic aspects depends on measuring the economic factors (real state value before and after proposed project, encouraging investment, economic impact on tourism development, providing job opportunities) as dependent factors according to the participant point of view as an independent factor. In the following assessment, we will discuss the result of the investigation. The main indicator result will be included in the research, and other indicators will be in the Appendix A (Tables A1–A18).

7.2.1. Real State Value before and after the Proposed Project

The results show that the research investigation sample accepts that the proposed resettlement project will increase the value of the real estate in historical Jeddah. This can appear from the variable results of Tables 4 and 5. The sample agreed that the increase would be for the heritage reuse buildings and all property in historical Jeddah.

**Table 5.** ANOVA table for previous factors.

| | | | Sum of Squares | df | Mean Square | F | Sig. |
|---|---|---|---|---|---|---|---|
| HB reuse and impact on E V for property participants | Between Groups | (Combined) | 0.175 | 2 | 0.088 | 0.116 | 0.891 |
| | Within Groups | | 47.643 | 63 | 0.756 | | |
| | Total | | 47.818 | 65 | | | |

7.2.2. Proposed Project and Encouraging Investment

The results show that the research investigation sample accepts that the proposed resettlement project will encourage investment in this area in reuse, service, and development projects. This appears from the result that the mean of the sample variable was 2.54, and the significance in ANOVA was 0.716, which was more than 0.05.

### 7.2.3. Proposed Project and Economic Impact on Tourism Development

The result shows that the research investigation sample accepts that the proposed project of resettlement will increase the development of tourism besides the development projects in Jeddah center, which will be integrated, the mean of the variable was 2.42. The significance in ANOVA was 0.944, which was more than 0.05.

### 7.2.4. Proposed Project and Providing Job Opportunities

The results show that the research investigation sample accepts that the proposed resettlement project will increase job opportunities in historical Jeddah in the fields of heritage restoration, heritage reuse, and touristic projects. The mean of the variable was 2.16, and the sig. in the ANOVA text was 0.773 more than 0.05.

### 7.3. *Assessment of Environmental Aspects*

The assessment of economic aspects depends on measuring the environmental factors (improvement of the visual image, control the levels of pollution-energy consumption, efficiency of the current cleaning and firefighting system) as dependent factors according to the participant point of view as an independent factor. In the following assessment, we will discuss the result of the investigation. Main indicator results will be included in the research, and other indicators will be in the Appendix A (Tables A1–A18).

### 7.3.1. Improvement of the Visual Image

The result shows that the research investigation sample accepts that the proposed resettlement project will improve the visual image as a result of the renovation heritage building and reuse project will be one of the sustainability factors; the mean of the variable was 1.74. The significance in ANOVA was 0.376, which was more than 0.05.

### 7.3.2. Control of the Levels of Pollution

The results show that the research investigation sample accepts that the proposed resettlement project will control the level of pollution as a result of preventing vehicle movement. Still, there is a note from that the investigated sample agrees with preventing car movement but agreed less with providing car parking in the building; the mean of the variable was 2.83. The significance in ANOVA was 0.735, which is more than 0.05 (Tables 6 and 7).

**Table 6.** The mean and standard deviation of reducing cars and using roads as a pedestrian pass.

| Participants | Mean | N | Std. Deviation |
|---|---|---|---|
| local community | 2.8333 | 42 | 1.26716 |
| municipality | 3.1250 | 8 | 1.55265 |
| specialist | 2.6875 | 16 | 1.19548 |
| Total | 2.8333 | 66 | 1.27199 |

**Table 7.** The ANOVA table for previous factors.

| | | Sum of Squares | df | Mean Square | F | Sig. |
|---|---|---|---|---|---|---|
| reducing cars increasing pedestrians participants | Between Groups (Combined) | 1.021 | 2 | 0.510 | 0.309 | 0.735 |
| | Within Groups | 104.146 | 63 | 1.653 | | |
| | Total | 105.167 | 65 | | | |

### 7.3.3. Energy Consumption Control

The result shows that the research investigation sample accepts that the proposed resettlement project will decrease energy consumption as a result of the reuse of heritage buildings that are compatible with the environment and weather. The mean of the variable was 2.5, and the significance in ANOVA was 0.771, which was more than 0.05.

### 7.3.4. Efficiency of the Current Cleaning and Firefighting System

The result shows that the research investigation sample accepts that the current cleaning and firefighting system will be insufficient for implementing the proposed project, and these systems must be reconsidered according to the new project. The mean of the variable was 1.7, and the significance in the ANOVA was 0.536, which was more than 0.05.

### 7.4. Analysis of the Investigation Results

Table 8 shows the matrix of the relationship between participants and indicators.

**Table 8.** Matrix of the relationship between participants and indicators.

| | | Local Community | Municipality | Specialists |
|---|---|---|---|---|
| Social aspect | Acceptance of living in heritage buildings | Accepting living in heritage building returns community appreciation for the heritage and pay attention to the historical Jeddah. | The municipality presents support to restoration projects and established historical festivals that support community acceptance. | The proposed project represents the solution to reuse this urban district, which helps specialists to support the project and sustainable development. |
| | Service efficiency | Services will be one of the attraction factors to the local community, which will help and sustain the proposed project. | A strategic plan should be prepared to raise the efficiency of services that are needed to sustain the conservation project. | Studding services should be integrated with the development project in Jeddah. |
| | Appropriateness of heritage building spaces to the target community | Spaces, such as for car parking and private gardens, should be included in the buildings. | Current spaces are enough for the target community, but the building is suitable for one family according to community privacy. | |
| | Urban space's appropriateness to the lifestyle of the target community | Current urban spaces are enough for the target community, which was evident during the Jeddah historical festival, but some activities should be included in these urban spaces, which will help placemaking. | | |
| | Relationship between the resettlement project and development projects | The project is considered integrated with development projects in the region; therefore, services must be studied within the framework of this integration to achieve sustainability. | | |
| Economic aspect | Real estate value before and after the proposed project | The project targeted classes that help raise the rental value and thus the value of the buildings. | The various uses) housing, services, touristic) that the project provides for the area helps to raise the real estate value. | |
| | Proposed project and encouraging investment | The project encourages investment at the famous and governmental levels, which helps to provide funding from multiple sources for projects. | | Diversity of projects and funding sources sustains conservation operations in the area. |
| | Proposed project and economic impact on tourism development | The project helps to develop the area and thus encourage tourism, which will provide, in addition to identifying the urban heritage of the area, getting to know customs and traditions of the local community. | | |
| | the proposed project and Providing job opportunities | The project provides many job opportunities, whether during conservation or reuse projects or from touristic projects that are produced from conservation projects. | | |

**Table 8.** *Cont.*

| | | Local Community | Municipality | Specialists |
|---|---|---|---|---|
| Environmental aspect | improvement of the visual image | Restoration and reuse of heritage buildings will improve the visual image, in addition to changing the local community according to the proposed project. | | |
| | Control the levels of pollution | Pollution will be decreased due to energy consumption reduction and using roads as pedestrian paths. | | |
| | energy consumption control | Using buildings that are adequate for the Jeddah environment will decrease energy consumption. | | |
| | The efficiency of the current cleaning and firefighting system | The current project should be developed to be adequate for the proposed project. | The current project is adequate to municipality strategy and UNESCO requirements, but it should be improved in case of implementing new projects. | |

## 8. Discussion

Resettlement of the Saudi community in the historical area of Jeddah is one of the sustainable solutions for the preservation of urban heritage in historical Jeddah. The research addresses the problem of the multiplicity of heritage buildings in the area, as the area represents an integrated heritage neighborhood, which makes it challenging to find the multiple functions of the buildings. The area is currently under development through many projects that make the area one of the most critical points of attraction in the city, which helps to restore the original community of the area to it. The resettlement project was represented in the rehabilitation of the Hafsia quarter in Tunis; the project achieved economic and social targets [29].

The research discussed the research hypothesis, which proposes reusing heritage buildings. The reuse of buildings is one of the most critical factors for the sustainability of preservation operations. Reuse represents the source of post-conservation economies, according to Elbelkasy's study [14], which ensures sustainable financing for maintenance operations. There is a link between conservation and the urban development of heritage areas, which works to preserve the surrounding environment and develop the local community. The research proposed the resettlement process to be adequate to the nature of the area, an integrated heritage neighborhood, which must be considered when reusing buildings so that residential use is provided that can be used in this large number of buildings. The resettlement project considers this specificity in terms of the ability to accommodate the most significant number of buildings in the proposed project and offer many development projects aimed at returning the city center to the area again. This area is an attractive element for the indigenous people who abandoned it in the past.

The questionnaire discussed the sustainability factors for the reuse of buildings according to the research hypothesis, which was divided into three levels: the level of society, the economy, and the environment, provided that the questionnaire addressed the opinion of the local community, local administration (municipality), and specialists. The questions at the community level first addressed the acceptance of the project from the ground up, and the sample accepted the resettlement project based on the results of the questionnaire. In reviewing the results, we found that the average of the local community indicates approval to some extent, which indicates the community's fear of the idea, despite its acceptance of it, which is a result of the nature of the community, in addition to the lack of services available in the area. There must be a clear plan to improve it because it can increase the community's acceptance of the project idea.

One of the essential factors that helps the local community to accept the idea of the project is the suitability of the spaces of the heritage buildings to the nature of the local community. Indeed, the spaces in the heritage buildings are suitable for residential use for which they were established. However, some spaces have become one of the needs of Saudi families, especially in the classes targeted for resettlement, for example, a place to park a car in the house, which contradicts the nature of the area, in which the pollution rates must be reduced by preventing the movement of motorized vehicles. The public spaces are commensurate with the nature of the target community, which was shown, for

example, during the Historic Jeddah Festival. The top-down adaptive reuse of the heritage area should help sustainable development according to the example of reuse of Xintiandi in the Luwan district Shanghai [30].

The reuse of buildings provides the economic resources needed for maintenance operations that support the sustainability of conservation operations. The research hypothesis deals with regional economies to provide the necessary restoration and maintenance operations due to using buildings for reuse and raising the real estate value of the buildings, and providing job opportunities, whether during the restoration and reuse period or after. Through the provision of services in the region. The sample agreed with the project's importance from an economic point of view. It is integrated with the development projects that were recently announced for the development of the central Jeddah region.

There is no doubt that heritage preservation operations are followed by the development of the local environment for heritage environments, and the reuse of heritage buildings improves the visual image of the area. The sample agreed that using buildings for resettlement would reduce pollution due to heritage buildings' compatibility with climatic factors and thus reduce energy consumption. Allocate streets for pedestrians and prevent the movement of vehicles to reduce pollution. Despite the sample agreeing with preventing vehicles, they were fearful of the lack of parking spaces in or near heritage buildings. Therefore, the target community's awareness should be increased and try to change its behavior, especially with the widespread use of smart transportation components [31].

The research discussed the research hypothesis that dealt with the reuse of heritage buildings in Historic Jeddah for resettlement, which achieves the sustainability of preservation operations of urban heritage in Historic Jeddah, which agreed with Akram Ljla study that heritage building reuse might generate sustainability [32]. Reusing heritage buildings in the case of historic Jeddah, the resettlement of the original community in the heritage area helps sustain conservation operations in the area. It works on developing the local community, strengthening the region's economies, raising its real estate value, and the positive effects on the urban environment in the region.

## 9. Results

The research results can be concluded as follows:

- Policies to preserve heritage areas that are concerned only with restoration and preservation are deficient and unsustainable.
- The local community is one of the most important guarantees for the success and sustainability of preservation operations, and heritage festivals were one of the most critical factors attracting the attention of the local community in Historic Jeddah, which helped raise the heritage awareness of the original community in the area of its importance.
- The residential use of heritage buildings is one of the uses that are commensurate with the nature of these buildings, as it is the original use of them, and it is commensurate with the reuse of heritage neighborhoods in which these buildings are numerous, as is the case of heritage Jeddah; however, the spaces in heritage building should be redesigned to be adequate to the target community.
- The reuse of heritage buildings in resettlement works to sustain the preservation operations of Historic Jeddah and integrates with the development projects that have been announced in the areas adjacent to the historic area.
- Reusing the heritage buildings in historic Jeddah increases the economic value of real estate and provides many job opportunities in the region, whether in restoration or services after the operation.
- Providing the necessary services to the target community is one of the essential factors of attraction and the success of the reemployment operations in the region, as it ensures the participation of the community and their transfer to housing in the region again, especially with the attractions provided by the new development projects that return the city center to the region again.

- Economic incentives are among the factors that attract the local community in the historical area of Jeddah to participate in the heritage building reuse project, especially with the privacy of the community and its adherence to heritage homes, which are inherited from generations. Renting it to others is one of the strange procedures for conservative societies, such as Saudi society.

*Recommendations*

- Implementing a pilot project through which the idea can be presented and evaluated helps the interaction of the local community.
- Raising awareness of the local community about the importance of the project and its impact on the sustainability of conservation operations in the area and the importance of participating in the project.
- Study current services and identify essential services needed for the target community to study how to integrate them into the heritage area.
- Study the environmental impact of the project and the expected service projects. The environmental impact of tourism projects in the region and the impact of tourism projects on the target community of resettlement operations and the region must be studied.
- Study the mutual impact between the proposed project and development projects in the region to develop an integrated vision for the region and not repeat services or projects, especially tourism and culture.
- Increase community awareness of creating a car-free historic area and making it pedestrian-friendly, especially after implementing the proposed project.
- Redesign the heritage buildings spaces and present them to the local community for their opinion.

## 10. Conclusions

The main research findings and interpretations are:

- Dealing with the reuse of buildings in the historic Jeddah area must accommodate many buildings. The most appropriate function that can accommodate this large number of buildings is to use them in their original functions.
- The peculiarity of the historical area of Jeddah, which helped it to be on the World Heritage List by UNESCO, makes the sustainability of the region's conservation operations one of the most important criteria that must be considered when developing policies for preserving the region.
- The local community of heritage areas is considered one of the essential elements for the sustainability of their preservation operations. The historical area of Jeddah has suffered from local community change and its replacement with a community of expatriates.
- The proposed development projects in the areas adjacent to the historical area restore the historical area once again like a magnet for the Saudi community, which helps accept the proposed project and provides the necessary services for it.
- Reusing the heritage buildings in restructuring the area's local community through resettlement achieves the sustainability of preservation operations in the area and increases the real estate value of the buildings and the area, which was verified through the applied study on the area, then the research hypothesis is achieved.
- The extent of the interaction of Saudi society with the region can be studied by studying the interaction between the community and the historical area of Jeddah through the heritage festivals that took place in the region and have helped introduce the new generations to the region and monitor their interactions with it, which helps to accept returning to the region again.

The research discussed the research hypothesis by studying the sustainability of historical preservation operations by measuring a set of indicators at the level of society, economy, and environment for the historical area, and these indicators were measured through

a questionnaire targeting stakeholders (community, local environmental administration, specialists) to reach the extent to which the proposed project achieves the objectives of the sustainability of conservation operations and thus proves or denies the research hypothesis, and the research has relied on the results of the sample that were affected by the limited research. As for the second field, it is the environmental impact assessment of the proposed project, along with the assessment of the environmental impact of development projects in the vicinity of the historical area, with the study of more indicators that help to reach more results or the implementation of a pilot project through which more results can be reached and the extent to which the community actually interacts with the proposal.

**Author Contributions:** Conceptualization, M.I.E.-b. and S.A.W.; Data curation, M.I.E.-b. and S.A.W.; Formal analysis, M.I.E.-b.; Investigation, M.I.E.-b.; Methodology, M.I.E.-b. and S.A.W.; Resources, M.I.E.-b. and S.A.W.; Supervision, M.I.E.-b.; Writing—review & editing, M.I.E.-b. All authors have read and agreed to the published version of the manuscript.

**Funding:** This research received no external funding.

**Institutional Review Board Statement:** Not applicable.

**Informed Consent Statement:** Not applicable.

**Conflicts of Interest:** The authors declare no conflict of interest.

## Appendix A

**Table A1.** The mean and standard deviation of service efficiency.

| Participants | Mean | N | Std. Deviation |
|---|---|---|---|
| local community | 3.7381 | 42 | 0.88509 |
| municipality | 3.8750 | 8 | 1.35620 |
| specialist | 3.5000 | 16 | 1.21106 |
| Total | 3.6970 | 66 | 1.02236 |

**Table A2.** ANOVA table for the previous factors.

| | | Sum of Squares | df | Mean Square | F | Sig. |
|---|---|---|---|---|---|---|
| services efficiency participants | Between Groups (Combined) | 0.945 | 2 | 0.473 | 0.444 | 0.643 |
| | Within Groups | 66.994 | 63 | 1.063 | | |
| | Total | 67.939 | 65 | | | |

**Table A3.** The mean and standard deviation of the appropriateness of heritage buildings spaces to the target community.

| Participants | Mean | N | Std. Deviation |
|---|---|---|---|
| local community | 3.4524 | 42 | 0.91605 |
| municipality | 3.5000 | 8 | 1.51186 |
| specialist | 3.6250 | 16 | 0.88506 |
| Total | 3.5000 | 66 | 0.98058 |

**Table A4.** ANOVA table for the previous factors.

| | | Sum of Squares | df | Mean Square | F | Sig. |
|---|---|---|---|---|---|---|
| HB appropriateness to T C participants | Between Groups (Combined) | 0.345 | 2 | 0.173 | 0.175 | 0.840 |
| | Within Groups | 62.155 | 63 | 0.987 | | |
| | Total | 62.500 | 65 | | | |

**Table A5.** The mean and standard deviation of the appropriateness of urban spaces to the target community.

| Participants | Mean | N | Std. Deviation |
|---|---|---|---|
| local community | 2.8333 | 42 | 1.05730 |
| municipality | 2.2500 | 8 | 0.70711 |
| specialist | 2.8125 | 16 | 0.75000 |
| Total | 2.7576 | 66 | 0.96174 |

**Table A6.** ANOVA table for the previous factors.

| | | Sum of Squares | df | Mean Square | F | Sig. |
|---|---|---|---|---|---|---|
| HU appropriateness to T C participants | Between Groups (Combined) | 2.350 | 2 | 1.175 | 1.282 | 0.285 |
| | Within Groups | 57.771 | 63 | 0.917 | | |
| | Total | 60.121 | 65 | | | |

**Table A7.** The mean and standard deviation of the relation between the proposed project and encouraging investment.

| Participants | Mean | N | Std. Deviation |
|---|---|---|---|
| local community | 2.5714 | 42 | 1.08522 |
| municipality | 2.2500 | 8 | 1.03510 |
| specialist | 2.6250 | 16 | 1.20416 |
| Total | 2.5455 | 66 | 1.09800 |

**Table A8.** ANOVA table for the previous factors.

| | | Sum of Squares | df | Mean Square | F | Sig. |
|---|---|---|---|---|---|---|
| HB reuse and P S investment participants | Between Groups (Combined) | 0.828 | 2 | 0.414 | 0.336 | 0.716 |
| | Within Groups | 77.536 | 63 | 1.231 | | |
| | Total | 78.364 | 65 | | | |

**Table A9.** The mean and std. deviation of proposed project impact on touristic development.

| Participants | Mean | N | Std. Deviation |
|---|---|---|---|
| local community | 2.4048 | 42 | 1.03734 |
| municipality | 2.3750 | 8 | 1.40789 |
| specialist | 2.5000 | 16 | 0.89443 |
| Total | 2.4242 | 66 | 1.03865 |

**Table A10.** ANOVA table for the previous factors.

| | | | Sum of Squares | df | Mean Square | F | Sig. |
|---|---|---|---|---|---|---|---|
| HB reuse and touristic D participants | Between Groups | (Combined) | 0.127 | 2 | 0.064 | 0.057 | 0.944 |
| | Within Groups | | 69.994 | 63 | 1.111 | | |
| | Total | | 70.121 | 65 | | | |

**Table A11.** The mean and standard deviation of the relation between the proposed project and job opportunities.

| Participants | Mean | N | Std. Deviation |
|---|---|---|---|
| local community | 2.1190 | 42 | 0.96783 |
| municipality | 2.1250 | 8 | 0.99103 |
| specialist | 2.3125 | 16 | 0.79320 |
| Total | 2.1667 | 66 | 0.92126 |

**Table A12.** ANOVA table for the previous factors.

| | | | Sum of Squares | df | Mean Square | F | Sig. |
|---|---|---|---|---|---|---|---|
| HB reuse and job opportunities participants | Between Groups | (Combined) | 0.449 | 2 | 0.225 | 0.259 | 0.773 |
| | Within Groups | | 54.717 | 63 | 0.869 | | |
| | Total | | 55.167 | 65 | | | |

**Table A13.** The mean and standard deviation of the relation between the proposed project and improving visual image.

| Participants | Mean | N | Std. Deviation |
|---|---|---|---|
| local community | 1.7143 | 42 | 0.80504 |
| municipality | 2.1250 | 8 | 1.24642 |
| specialist | 1.6250 | 16 | 0.71880 |
| Total | 1.7424 | 66 | 0.84691 |

**Table A14.** ANOVA table for the previous factors.

| | | | Sum of Squares | df | Mean Square | F | Sig. |
|---|---|---|---|---|---|---|---|
| HB reuse and improving V I participants | Between Groups | (Combined) | 1.425 | 2 | 0.712 | 0.993 | 0.376 |
| | Within Groups | | 45.196 | 63 | 0.717 | | |
| | Total | | 46.621 | 65 | | | |

**Table A15.** The mean and std. deviation of the relation between the proposed project and decrease of energy consumption.

| Participants | Mean | N | Std. Deviation |
|---|---|---|---|
| local community | 2.4286 | 42 | 1.19231 |
| municipality | 2.7500 | 8 | 1.66905 |
| specialist | 2.5625 | 16 | 1.03078 |
| Total | 2.5000 | 66 | 1.20576 |

**Table A16.** ANOVA table for the previous factors.

| | | Sum of Squares | df | Mean Square | F | Sig. |
|---|---|---|---|---|---|---|
| HB reuse and decreasing E P participants | Between Groups (Combined) | 0.777 | 2 | 0.388 | 0.261 | 0.771 |
| | Within Groups | 93.723 | 63 | 1.488 | | |
| | Total | 94.500 | 65 | | | |

**Table A17.** The mean and standard deviation of the efficiency of the current cleaning and firefighting system.

| Participants | Mean | N | Std. Deviation |
|---|---|---|---|
| local community | 1.6905 | 42 | 0.74860 |
| municipality | 1.7500 | 8 | 0.88641 |
| specialist | 1.9375 | 16 | 0.68007 |
| Total | 1.7576 | 66 | 0.74546 |

**Table A18.** ANOVA table for the previous factors.

| | | Sum of Squares | df | Mean Square | F | Sig. |
|---|---|---|---|---|---|---|
| HB reuse and developing clean F F projects participants | Between Groups (Combined) | 0.708 | 2 | 0.354 | 0.629 | 0.536 |
| | Within Groups | 35.414 | 63 | 0.562 | | |
| | Total | 36.121 | 65 | | | |

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
