# Peer review of "Sustainable Conservation and Reuse of Historical City Center Applied Study on Jeddah—Saudi Arabia"

_sustainability, doi:10.3390/su14095188_

Round 1
Reviewer 1 Report
Dear Authors,
The research topic about " Sustainable Conservation and Re-Use of Historical City Center Applied Study on Jeddah – Saudi Arabia” discussed in the article is currently an important topic, due to the fact that cities play and preservation of heritage areas a pivotal role in social and economic aspects around the world and have huge impact on the environment. The research aims to study the proposal to resettle the local community of Historic Jeddah and to return it to the region by studying the impact of this proposal on the sustainability of conservation operations in the Jeddah heritage area and the extent of the proposal’s compatibility with development projects in the region.
After reading the paper, I have comments and suggestions to improve the paper as follows:
Introduction
This part of the article has been done correctly. The authors refer to world literature and present the contemporary meaning of "conservation in historical city". Emphasize that , in the past two decades, the concept of "historical city" has become more and more popular. Sustainability of the heritage conservation in historical city center depends on the reuse of heritage buildings and revival of historical center's urban fabrics; therefore, a rehabilitation plan should be occurring to use all urban fabric of this historical center Cities play a pivotal role in social and economic aspects around the world and have a huge impact on the environmen .
Materials and Methods
According to the Sustainability Journal guidelines there should be Materials and Methods chapter. I suggest including subsections 1.3 Research Methodology to this chapter and describing the steps in more detail. The diagram alone is not enough. however, there are no specific research stages and a description of the methods that were used in the work.
Results
Suggests correcting technical errors. This chapter contains too many subsections and numbers.
were presented and described in a very good manner and are very interesting. They contribute to the value of this paper. No changes are required.
In the Discussion Section, the authors should discuss and explain the findings and results of the paper more. It also important to describe the results of the paper in greater detail in this section. This would contribute to a high improvement of this paper. The authors should compare their project and results with results from similar conducted research on this topic from other parts of Asia and all around the world.
In References you should mind correct spelling and adjustments according to Sustainability Guidelines.
There are errors such as: Spaces are missing in this references.
[18, 21]
Author Response
Dear respected reviewer
Thank you for your valuable notes, which will improve the research. The attached file is the modified file according to your notes.
Yours
authors

Reviewer 2 Report
- In the Introduction section, please find another expression to refer “city center”, due to many repetitions of this expression, specially in the first paragraph.
- Please, review the English orthography, including the capital letters at the beginning of each title, phrase, tables or caption
- Please specify the conservation strategy underneath the Historic Jeddah restauration process, considering a sustainable approach.
- Please use the same number of significant figures for the mean and standard deviation
Author Response

(The authors gave the same response as above.)
